# Biomarkers in IBD: What to Utilize for the Diagnosis?

**DOI:** 10.3390/diagnostics13182931

**Published:** 2023-09-13

**Authors:** Renata D’Incà, Giulia Sturniolo

**Affiliations:** 1Department of Surgical, Oncological and Gastroenterological Sciences, University of Padua, 35124 Padua, Italy; 2Department of Women’s and Children’s Health, University of Padua, 35128 Padova, Italy

**Keywords:** fecal calprotectin, C-reactive protein, p-ANCA, ASCA

## Abstract

The role of biomarkers in the diagnosis of inflammatory bowel disease is not fully characterized. C-reactive protein has a short half-life and elevates quickly after the onset of an inflammatory process; the performance is better in Crohn’s disease than in ulcerative colitis. Erythrocyte sedimentation rate is easy to determine, widely available, and cheap, but the long half-life, the influence of age, anemia, smoking, and drugs limit its usefulness. Fecal markers have good specificity, but suboptimal accuracy. Microbial antibodies and novel immunological markers show promise but need further evidence before entering clinical practice. Proteomic methods could represent the dawn of a new era of stool protein/peptide biomarker panels able to select patients at risk of inflammatory bowel disease.

## 1. Introduction

Inflammatory bowel disease (IBD) diagnosis is currently based on clinical criteria together with biochemical and instrumental investigations. A prompt diagnosis is advisable in order to offer restitution to well-being and good quality of life to the patient. The recognition and diagnosis of IBD and its differential diagnosis from other acute and chronic bowel diseases is crucial for offering proper treatment and good prognosis. IBD therapy should induce a rapid symptom control, normalize biochemical indexes and resolve endoscopic lesions. With these targets in mind, we can hope to clear the disease or at least to slow the progression, to reduce the need for steroids and to avoid complications and surgeries in the patients.

## 2. Serological Markers

### 2.1. Routine Blood Tests

If IBD is suspected, laboratory tests can guide further investigation and contribute to the differential diagnosis. Blood count, electrolytes, erythrocyte sedimentation rate (ESR), C-reactive protein (CRP) and stool cultures are routine exams which can be easily performed on peripheral blood samples and are useful to detect acute ongoing inflammation. However, they are not specific to detect intestinal inflammation. Serum and fecal inflammation markers that reflect the intestinal damage might be useful both in terms of diagnosing the disease and for the management and prognosis. A list of the most studied parameters is reported in Table 1.

Table 2 summarizes the performance of serological and fecal tests in the diagnosis of IBD.

Especially when managing pediatric patients, non-invasive investigations should represent first-line assessment. The performance of blood tests has been tested in pediatric patients with clinically suspected IBD; two thirds of them confirmed the IBD diagnosis [2]. Platelet count and hemoglobin rate were found the most reliable tests for differentiating IBD from non-IBD patients (*p* < 0.002 and *p* < 0.007, respectively). Anemia or thrombocytosis had a diagnostic sensitivity and specificity of 91% and 80%, with a positive predictive value of 94% and a negative predictive value of 76%.

Anemia is frequent in IBD. It may stem from multiple sources, the most obvious being acute, chronic, or occult bleeding, but the role of chronic inflammation also needs to be considered [3,4]. The biochemical assessment of iron metabolism allows discrimination of iron-deficiency anemia which is characterized by a low serum iron and ferritin with high serum levels of transferrin and total iron binding capacity, from anemia induced by chronic inflammation which has a normal/high ferritin, low transferrin, and total iron binding capacity.

Thromboembolic disease has been described in IBD, and especially in ulcerative colitis, possibly secondary to colonic inflammation. During active disease phases [5], not only inflammatory parameters (CRP, ESR, platelet count) but also coagulation parameters (thrombin–antithrombin complexes, fibrinogen, FgDP, and FbDP) increase, the coagulation and fibrinolytic cascades are activated in inflamed ulcerative colitis, with the hemostatic arm favoring coagulation. This hypercoagulable state may persist also when the acute phase subsides.

ESR has been found related to the clinical activity and extent of disease in ulcerative colitis [6]. The ability of clinical and biochemical markers to predict failure of medical therapy has also been tested in 67 patients admitted for severe colitis. ESR > 75 mm/h together with fever > 38 °C on the day of arrival were selected as the best predictors of the need for a colectomy [7].

CRP is inexpensive and easy to perform. CRP is synthesized by hepatocytes following the release of IL-6, TNF alfa and IL 1 beta. The molecular structure consists of five subunits that make up a pentameric appearance. Its production is quite rapid and its short half-life (18 h) makes the probe useful to rapidly detect inflammatory changes and to follow up the process [8]. It rapidly declines with effective therapy and represents a useful prognosticator in ulcerative colitis. More than eight bowel movements a day and a CRP higher than 45 mg/L after 3 days of intensive medical therapy carry an 85% chance of colectomy in patients with severe colitis [8].

However, there is considerable heterogeneity in CRP generation based on the genetics of an individual patient [9]. CRP is a poor parameter of inflammation, especially in ulcerative colitis; about 50% of ulcerative colitis patients have normal CRP during a disease flare [10]. Moreover, elevated serum CRP is not unique for intestinal inflammation, occurring in response to most systemic inflammatory disorders.

Recently, several inflammation-based ratios of different biomarkers have been proposed in IBD diagnosis and prediction of IBD. Several different ratios demonstrated significant differences between IBD and healthy controls: a retrospective analysis found increased levels of the neutrophil-to-albumin ratio, neutrophil-to-pre-albumin ratio, fibrinogen-to-pre-albumin ratio (*p* < 0.001), and lower levels of albumin-to-alkaline-phosphatase ratio, as well as the Prognostic Nutritional Index (*p* < 0.001) [11].

### 2.2. Immunological and Other Antibody Markers

p-ANCAs (perinuclear anti-neutrophil cytoplasmic antibodies) have been described in ulcerative colitis patients for thirty years, though the exact epitope remains unknown [12,13]. Examined by indirect immunofluorescence, the neutrophil labeling pattern of these antibodies shows a perinuclear staining distinguishable from the one produced by ANCA in Wegener granulomatosis, which exhibits a diffuse fluorescent neutrophil cytoplasm labeling. P-ANCAs have a sensitivity of 52% and a specificity of 91% in differentiating ulcerative colitis from Crohn’s disease [14].

Proteinase 3 antineutrophil cytoplasmic antibody (PR3-ANCA) is a serologic marker for granulomatosis with polyangiitis. Its role as a diagnostic marker for ulcerative colitis was assessed in 173 ulcerative colitis patients including 77 patients with new-onset ulcerative colitis (disease diagnosed within 1 month), 110 Crohn’s disease patients, 48 patients with other gastrointestinal diseases and 71 healthy controls. PR3-ANCA ≥ 3.5 U/mL demonstrated a 44.5% sensitivity and a 95.6% specificity for the diagnosis of ulcerative colitis. PR3-ANCA positivity was more prevalent in the 77 new-onset ulcerative colitis patients (58.4%). PR3-ANCA measurement has proved to be useful not only for diagnosing ulcerative colitis but also for evaluating disease severity and extension and predicting the clinical course [15].

Recently, serum IgG anti-integrin αvβ6 autoantibodies (IgG anti-αvβ6) have been described as highly sensitive and specific serological markers of ulcerative colitis in the sera of Japanese IBD patients [16], and the association was confirmed in Sweden [17].

Anti-Saccharomyces cerevisiae antibodies (ASCA) are antibodies directed against the cell wall of baker’s yeast. This antibody occurs in the serum in Crohn’s disease, but not in ulcerative colitis. In a meta-analysis on 14 studies performed in Europe, Israel and Canada [18], serum levels of the ASCA had a sensitivity of 56% and a specificity of 88% in discriminating Crohn’s disease from ulcerative colitis. Moreover, ASCA-positive Crohn’s disease patients have a higher risk of disabling disease, including stricturing and penetrating behavior, earlier onset and perianal disease, resulting in an increased need for surgery [18,19]. Both p-ANCA and ASCA are highly specific, but their sensitivity is rather low, so these tests remain unsuitable for screening purposes [20]. Sensitivity and specificity of ASCA+/p-ANCA− are reported to be higher (0.70 and 0.93, respectively) in pediatric Crohn’s disease [21,22]. Overall, their accuracy in diagnosis of IBD is suboptimal, ranging between 71 and 80% [23]. p-ANCA and ASCA have been documented in serum even years before the diagnosis of IBD. The early detection of these markers might therefore help identify the risk of future disease and pinpoint patients with a more aggressive clinical course [24].

Other anti-glycan antibodies, namely anti-chitobioside carbohydrate IgA (ACCA), anti-mannobioside carbohydrate IgG (AMCA), anti-laminaribioside IgG (ALCA), anti-laminarin carbohydrate antibodies (anti-L), and anti-chitin carbohydrate (anti-C), have been detected in the serum of IBD patients. Their levels remain stable over time [25]. In general, each antiglycan antibody shows good specificity and positive predictive value in discriminating Crohn’s disease from ulcerative colitis [25].

The usefulness of a panel of anti-glycan antibodies may increase sensitivity as demonstrated in a small Israelian cohort of ASCA negative Crohn’s disease patients who proved positive to ALCA or ACCA in 44% of the patients, reaching sensitivity and specificity of 77 and 90%, respectively [18].

High similarity in anti-OmpC and anti-Pseudomonas fluorescence-associated peptide I2 antibodies has been demonstrated in discordant monozygotic twins. Twin pairs, but not discordant dizygotic twin pairs, suggest that both Anti-OmpC and anti-I2 stand for a genetically determined loss of tolerance [26].

In addition, positivity for ASCA IgA, ASCA IgG, anti-OmpC, anti-CBir1, or anti-flagellin antibodies was detected in 65% of the Crohn’s disease patients six years before diagnosis [27]. The same degree of prediction of IBD diagnosis has been confirmed with a panel of serum antibodies, including p-ANCA, ASCA IgA, ASCA IgG, anti-OmpC, and anti-CBir1, detected in samples stored four years previously [28].

The role of anti-glycan antibodies in disease pathogenesis has also been investigated in unaffected first-degree relatives. Both qualitative and quantitative analysis revealed that unaffected first-degree relatives have increased antibody response to microbial antigens. This impaired immunological response towards enteric microorganisms may result from genetic predisposition [29].

Recently, new evidence on the predictive role of microbial antibodies and immune-inflammatory markers has been explored using archive serum samples from the US Defense Medical Surveillance System [30]. A panel of inflammatory proteins and markers involved in cytokine signaling, innate immunity and response to bacteria predicted the diagnosis of Crohn’s disease within 5 years with high accuracy. If this holds true, preventive microbiome-targeted interventions could be implemented in predisposed individuals. By contrast, this was not true for ulcerative colitis with p-ANCA being a suboptimal predictor of developing the disease [30].

New insights into the pathogenesis of IBD involve an inappropriate and persistent inflammatory response to commensal gut microbiota in genetically susceptible individuals. Indeed, studies show that the intestinal microbiota in IBD patients is distinct from that found in healthy subjects and bacteria play an important role in the onset and perpetuation of the inflammatory process. Microbial biomarkers hold promise in assessing disease activity, treatment effectiveness and in personalizing treatment strategies. Moreover, the potential benefits of microbiome-modulating interventions with the use of probiotics, prebiotics, antibiotics, and even fecal microbiota transplantation have reached the IBD field. IBD patients show multiple differences in the composition of gut microbiota with respect to healthy individuals, particularly regarding microbial diversity and relative abundance of specific bacteria. Patients with IBD show higher levels of Proteobacteria and lower amounts of Bacteroides, Eubacterium, and *Faecalibacterium* than healthy individuals [31].

In Crohn’s disease, a loss of beneficial bacteria, such as *Faecalibacterium prausnitzii* and an increase in *Escherichia coli*, have been observed. Particular strains of *E. coli*, such as enteroadherent *E.coli*, may be associated with disease in a subset of Crohn’s disease patients with ileal involvement [32].

The composition of the fecal microbiota has been less characterized in ulcerative colitis patients, but a lower abundance of *Roseburia hominis* (*p* < 0.0001) and *Faecalibacterium prausnitzii* (*p* < 0.0001) was found in ulcerative colitis patients compared to controls by real-time PCR analysis [31]. Longitudinal studies involving a large cohort of European IBD patients confirmed greater dysbiosis and lower microbial diversity in Crohn’s than in ulcerative colitis patients [33].

The important role that intestinal microbiota play in IBD pathogenesis has been recently investigated by means of serological profiling of 100 Crohn’s disease patients, 100 ulcerative colitis patients and 100 healthy controls against 1173 bacterial and 397 viral proteins from 50 bacteria and 33 viruses on protein microarrays. Anti-bacterial antibody responses showed greater differential prevalence among the three subject groups than anti-viral antibody responses. Novel antibodies against the antigens of *Bacteroidetes vulgatus* (BVU_0562) and *Streptococcus pneumoniae* (SP_1992) showed higher prevalence in CD patients relative to healthy controls, while antibodies against the antigen of *Streptococcus pyogenes* (SPy_2009) showed higher prevalence in healthy controls relative to ulcerative colitis patients. Using these novel antibodies, a biomarker panel was built to distinguish between Crohn’s disease vs. control, ulcerative colitis vs. control, and Crohn’s disease vs. ulcerative colitis, respectively [34].

The microbial signature specific to Crohn’s disease combined with either imaging techniques or fecal calprotectin data is proposed in decision-making when the diagnosis is initially uncertain.

### 2.3. Cytokines

In the inflamed mucosa, immune cell recruitment produces cytokines and this leads to a stimulation and amplification of the inflammatory cascade and some of them are now the preferred target of biological drugs. They may contribute to our understanding of the inflammatory events and features of IBD, but the increased expression of proinflammatory cytokines in the intestinal mucosa is not always accompanied by increased concentrations of cytokines in the circulation.

The levels of IL-33, a member of the IL-1 family, and its ligand-related protein, ST2, were increased in the mucosa as well as in the serum of ulcerative colitis patients, and a good correlation was found with disease severity [35].

Similar results were also obtained by Diaz-Jimenez et al., who investigated serum as well as intestinal ST2 levels in ulcerative colitis, Crohn’s disease and non-IBD patients. Patients with IBD (either Crohn’s disease or ulcerative colitis) showed significantly increased serum ST2 levels compared to healthy controls and non-IBD patients. Furthermore, the analysis focused on the group of ulcerative colitis patients which was more numerous, and significant differences were found with respect to endoscopic disease activity [36]. The possible role of IL-33 and IL1RL1 genetic polymorphisms in contributing to the risk of IBD has been investigated in a large cohort of Italian patients. Two different polymorphisms were found to contribute to the risk and the IL-33 rs3939286 was increased in frequency in patients with extensive ulcerative colitis [37].

Oncostatin M, an IL6 family cytokine quickly released during degranulation, has been implicated in the pathogenesis of IBD and as an emerging marker for non-responsiveness to anti TNF alfa therapy. Serum Oncostatin M looks promising because increased levels have been found in first-degree relatives of IBD families, in newly diagnosed patients and in patients with recurrent disease after surgery [38].

IL-10 is regarded as one of the anti-inflammatory cytokines associated with ulcerative colitis. This was first demonstrated in IL-10 knock-out mice, which develop spontaneous enterocolitis [39]. The IL-10 cytokine maintains the intestinal homeostasis by inhibiting the immune response during inflammation.

In the meta-analysis by Meng et al., eight human studies were found to examine the relationship between serum IL-10 level and IBD. The IL-10 levels in the serum samples of approximately 300 patients were moderately but significantly increased compared to healthy controls (*p* = 0.022) [40].

The measurement of circulating cytokines has received much attention in the past decades; however, the inconsistency of the results does not recommend their use outside research. This might be due to the local effect of some cytokines, to different conditions of sample storage or to the different type of assay.

### 2.4. MicroRNA (miRNA)

MiRNAs are a group of small noncoding RNAs, ~18–22 nucleotides, which act as regulators for post-transcriptional gene expression. The miRNAs circulate in the human peripheral blood but can also be found in urine, saliva, and feces [41].

The miRNAs are engaged in disease origin and development, and some are pathology specific [42].

MiRNAs affect the intestinal barrier and inflammatory reactions, so most recent research in the IBD field has measured circulating miRNAs in body fluids such as blood or feces and in homogenized tissue biopsies using microarray profiling techniques, quantitative reverse transcription PCR, and next-generation sequencing [43].

Even though many miRNAs are reported, MiR-21 and miR-155 have repeatedly been identified and seem to be the most related to IBD [44,45].

MiR-21, which increases the paracellular permeability of the intestinal epithelium and the level of TNF alfa, is possibly the most intriguing, with associations between miR-21 and IBD replicated in several studies [46,47].

Another study found that serum samples from IBD patients showed higher levels of miR-16, miR-21, and miR-223 than controls [48].

Paraskevi et al. reported increased levels of miR-16, miR-23a, miR-29a, miR-106a, miR-107, miR-126, miR-191, miR-199a-5p, miR-200c, miR-362-3p, and miR-532-3p in the blood of Crohn’s disease patients compared to that of healthy controls and elevated levels of miR-16, miR-21, miR-28-5p, miR-151-5p, miR-155, and miR-199a-5p in the blood of ulcerative colitis patients compared to that of controls [49].

### 2.5. Other Markers

Extraintestinal manifestations, as well as osteo-articular, cutaneous, ocular, hepatic, pancreatic, nephrological, endocrinological, hematological, pulmonary, and thromboembolic ones are well recognized as being associated with IBD.

Arthropathy represents the most frequent extraintestinal manifestation of IBD, reported in 10–35% of patients [50]. Biochemical markers are not useful for the diagnosis when the two different inflammatory events coexist. Serum human cartilage glycoprotein 39 (YKL-40) was investigated in IBD patients with articular symptoms. Serum levels of HC gp39 were significantly increased in IBD patients with versus those without arthropathy, or controls (*p* < 0.001 and *p* < 0.01, respectively). The protein level also seems to correlate with the number of joints involved, suggesting that this substance could be used as a disease activity marker in arthritis associated with IBD [51].

Hepatobiliary diseases in IBD may range from abnormal liver function tests to primary sclerosing cholangitis. Altered liver function tests have been reported in 11% of a large cohort of Swedish patients, especially related to intestinal inflammation, and was usually reversible after the disease was brought under control [52]. The occurrence of liver steatosis and increased transaminases has also been demonstrated in a multicenter study involving IBD patients and controls [53].

Primary sclerosing cholangitis has a prevalence in ulcerative colitis ranging from 2.5 to 7.5%. 82% of PSC patients are p-ANCA positive [54]. Most patients are asymptomatic at the time of diagnosis, and routine tests show increased alkaline phosphatase and gamma–glutamyl transferase.

Intestinal permeability reflects the integrity of the intestinal mucosal barrier, which enables the passage of luminal substances by unmediated diffusion [55]. Intestinal permeability can be assessed non-invasively in vivo by measuring the urinary excretion of orally administered sugars such as lactulose/mannitol, glucose and sucralose or radioactive probes such as 51Cr-EDTA. An increased lactulose/mannitol ratio, together with increased CRP levels, were independent predictors of a final diagnosis of small bowel disease in 261 consecutive patients referred to a tertiary referral center with chronic diarrhea [56]. In Crohn’s disease abnormal permeability is detected in 95% of patients with small bowel Crohn’s disease, while in Crohn’s colitis, the sensitivity is lower [57]. However, intestinal permeability has not gained widespread application as a screening test to discriminate between patients with Crohn’s disease and irritable bowel syndrome. The reason for this is probably that the urinary sugar analysis is time consuming and demanding, and there may be some concern that the tests lack specificity being abnormal in a variety of small intestinal diseases, such as celiac disease, acute gastroenteritis, food intolerance, and allergy [58].

## 3. Fecal Markers

### 3.1. Calprotectin and Lactoferrin

The presence of intestinal inflammation increases mucosal permeability, resulting in more leukocytes passing through the mucosa and penetrating into the intestinal lumen. Leukocytes can be retrieved in stools and detected under the microscope, but since their degranulation is quick, only fresh stools can be analyzed [59,60]. Some leukocyte proteins (such as lactoferrin and calprotectin) are durable, however, and can be used as surrogate markers of leukocyte activity.

Calprotectin is a calcium and zinc-binding protein of the S-100 protein family, which comprises 60% of the cytosolic protein in human neutrophils, and lactoferrin is a component of the granules of neutrophilic granulocytes [61]. Lactoferrin, like other neutrophil proteins such as elastase, myeloperoxidase and lysozyme, increases in concentration during the active phases of the disease by comparison with the periods of remission. Lactoferrin is stable, and its extracellular release is the most efficient [62]. Calprotectin and lactoferrin show high sensitivity and specificity for the presence of macroscopic inflammation [61].

Calprotectin levels can help differentiate between inflammatory and non-inflammatory bowel conditions such as diverticulosis and irritable bowel syndrome. In a prospective study of 870 consecutive patients referred for colonoscopy, elevated calprotectin levels (>50 mg/dL) were detected in 85% of patients with colorectal cancer, and 81% of those with inflammatory conditions, but also in 37% of patients with normal or trivial endoscopic findings. In patients referred for chronic diarrhea, sensitivity and negative predictive value were 100% in detecting either inflammation or cancer [63].

In gastrointestinal infections, especially of bacterial origin, fecal calprotectin concentrations are strongly elevated and correlate with disease severity [64,65].

Viral infections, including the coronavirus disease induced by SARS-CoV2, show abnormal calprotectin levels, although less elevated than in IBD and bacterial infections [66].

Since optimal calprotectin cut-offs are not established, clinicians may be challenged in the interpretation of intermediate concentrations of 150–250 µg/g (declared as a grey zone by STRIDE-II recommendations) [67]. On the contrary, values higher than 250 should prompt further evaluation such as endoscopy.

A meta-analysis on the utility of CPR, ESR, fecal calprotectin, and fecal lactoferrin showed that there was a very low probability of having IBD when CRP or fecal calprotectin were within the normal range [68].

Eight studies investigated the role of fecal calprotectin as a first-line investigation in pediatric patients with suspected IBD: the pooled sensitivity and specificity for the diagnostic utility of fecal calprotectin were 0.978 (95% confidence interval (CI), 0.947–0.996) and 0.682 (95% CI, 0.502–0.863), respectively; the positive and negative likelihood ratios were both very interesting [69].

Other conditions, such as necrotizing enterocolitis [70], graft-versus-host disease [71], and drug-induced enteropathy should be kept in mind [72].

Evidence of increased fecal calprotectin levels in predicting colorectal inflammation has also been found in adult patients referred for colonoscopy because of chronic diarrhea [73].

Calprotectin is now considered a biomarker for inflammation in the gastrointestinal mucosa with implications for clinical decisions, but its biology in the gut needs further studies. In the healthy mucosa, calprotectin has a broad spectrum of immunomodulatory properties which may drive the generation of reactive oxygen species during gut injury. Therefore, not only increased fecal calprotectin concentrations are a landmark of neutrophilic inflammation, but also gut inflammation induces epithelial calprotectin expression and secretion. Different cut-off values are suggested for different patient categories, i.e., higher for patients with known inflammatory conditions and lower for screening purposes. The diagnostic performances of non-invasive tests for IBD have been recently analyzed in an umbrella review [1]. The clinical scenarios included diagnosis of IBD vs. non-IBD, IBD vs. irritable bowel syndrome, IBD vs. functional gastrointestinal disorders, Crohn’s disease vs. ulcerative colitis. Fecal calprotectin and fecal lactoferrin proved to be the most sensitive (0.97 and 0.94, respectively) for distinguishing IBD from non-IBD. However, anti-neutrophil cytoplasmic antibodies (ANCA) and fecal lactoferrin were the most specific for it. Fecal calprotectin and fecal lactoferrin were the most sensitive and specific tests, respectively, to distinguish IBD from irritable bowel syndrome. All tests had low sensitivity for distinguishing Crohn’s disease from ulcerative colitis, with ASCA IgA having the highest specificity [1].

Calprotectin is now widely used in clinical settings involving IBD patients; nevertheless, we must keep in mind that the performance of this test is far from ideal. Moreover, the cut-off value for discriminating functional from organic bowel disease is not standardized and depends on the test used and type of assay [74].

The appropriate clinical use of fecal calprotectin might be influenced by the type of assay. Different optimal thresholds need to be settled depending on the type of assay (120 μg/g for ELISA, 50 μg/g for CLIA and 100 μg/g for turbidimetry). Moreover, within- and between-subject variability must be taken into account: sensitivity is satisfactory in distinguishing between controls and IBD patients in patients < 65 years, but it is lower in older patients (ROC area: 0.584; 95% CI: 0.399–0.769) [75].

### 3.2. HMGB1

The nuclear protein High-Mobility Group Box 1 (HMGB1) is actively secreted from immune cells in the extracellular matrix, where it behaves as a proinflammatory cytokine. HMGB1 was measured in stools of 40 IBD pediatric patients and 13 controls. HMGB1 protein levels were significantly increased (*p* < 0.001) in the stools of patients, but were undetectable in the controls; fecal HMGB1correlated well with fecal calprotectin levels (r: 0.77 in CD, r: 0.70 in UC; *p* < 0.01) [76].

The reliability of fecal HMGB1 compared with fecal calprotectin in detecting intestinal inflammation has been measured in pediatric and adult IBD patients. Fecal HMGB1 expression was significantly increased in pediatric and adult patients with Crohn’s disease and ulcerative colitis and correlated with disease severity fecal calprotectin and HMGB1 significantly correlated in pediatric and adult IBD patients. Moreover, in patients with clinical and endoscopic remission, only fecal HMGB1 showed a strong match with the degree of histological scores of inflammation [77].

### 3.3. S100B

S100 proteins have been demonstrated to exert a protective role in the gastrointestinal tract. S100B is typically expressed by enteroglial cells and can be detected in feces. Its role as a non-invasive indicator of gastrointestinal inflammation has been tested prospectively in 48 IBD patients. Unlike calprotectin, S100B was significantly decreased in IBD patients compared to non IBD-patients. At the onset of disease, the lowest levels were found, suggesting that S100B in feces could have a potential diagnostic value for IBD [78].

### 3.4. MiRNAs

MiRNAs can also be found in feces [79]; significant miRNA expression changes were observed in IBD patients for all studied miRNAs with the highest expression of miR-155 and miR-223 in testing and validation cohorts. miR-21, miR-155, and miR-223 display significant levels and could potentially be considered biomarkers for IBD [48].

### 3.5. Novel Markers

The search of new stool protein/peptide biomarkers for diagnosing IBD has been performed with novel proteomic methods: MALDI-TOF/MS (*m*/*z* 1000–4000) analysis for peptides and LTQ-Orbitrap XL MS analysis for proteins have shown interesting differences between IBD patients and controls [80]. The MALDI-TOF/MS spectra showed numerous features in IBD patients, unlike controls. Overall, 426 features (67 control-associated, 359 IBD-associated) were identified. In the exploratory cohort, the sensitivity and specificity of spectra classified as control or IBD (absence or presence of IBD-associated features) were 81% and 97%, respectively. Blind analysis of the validation cohort confirmed a 97% specificity, with a lower sensitivity (55%) paralleling active disease frequency. Following binary logistic regression analysis, IBD was independently correlated with MALDI-TOF/MS spectra (*p* < 0.0001), outperforming fecal calprotectin measurements (*p* = 0.029). IBD-associated over-expressed proteins included immunoglobulins and neutrophil proteins.

Recently, proteomic analysis was performed in stools of Crohn’s disease patients and controls by difference gel electrophoresis 2-DIGE and MALDI-TOF/TOF MS which were able to select three novel fecal biomarkers of gut inflammation that display good specificity and sensitivity for identifying IBD and significantly correlate with IBD severity [81].

## 4. Conclusions

Traditional biochemical tests remain helpful in guiding strategies for the diagnosis of IBD. Fecal calprotectin determination is useful to rule out the presence of intestinal inflammation and to avoid unnecessary invasive procedures. New potential indices are promising, but at the moment, the accuracy for diagnosing ulcerative colitis or Crohn’s disease is suboptimal and not ready for use in clinical practice.

The influence of disease location, symptom duration, type of therapy, the use of concomitant drugs and the baseline inflammation all contribute to the variability of the results. The uniqueness of each patient will probably be targeted in further studies by a panel of markers rather than one marker alone.

## Figures and Tables

**Table 1 diagnostics-13-02931-t001:** Noninvasive serological and fecal markers in IBD diagnosis.

Serological Markers	Fecal Markers
CRP	Calprotectin
ESR	Lactoferrin
Whole blood count: RBC, WBC, PLT	HMGB1
Ferritin	S100B
Anti-glycan antibodies, pANCA	MicroRNAs
Cytokines: IL 1, IL 10, Oncostatin M	
MicroRNAs	

**Table 2 diagnostics-13-02931-t002:** Diagnostic performance of serological and fecal tests in clinical situations suggesting IBD (from Shi et al., [1] modified).

Diagnostic Performance (95% CI)		
	Sensitivity	Specificity
Diagnosis: IBD vs. non-IBD		
Fecal calprotectin	0.88 (0.83–0.92)	0.80 (0.69–0.87)
CRP	0.63 (0.51–0.73)	0.88 (0.80–0.93)
ASCA	0.40 (0.38–0.42)	0.92 (0.91–0.94)
pANCA	0.33 (0.31–0.34)	0.97 (0.96–0.98)
Fecal lactoferrin	0.82 (0.72–0.89)	0.95 (0.88–0.98)
microRNAs	0.80 (0.79–0.82)	0.84 (0.82–0.86)
Diagnosis: IBD vs. Irritable Bowel Syndrome		
Fecal calprotectin	0.97 (0.91–0.99)	0.76 (0.66–0.84)
Fecal lactoferrin	0.78 (0.75–0.82)	0.94 (0.91–0.96)

## Data Availability

Not applicable.

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
