# Peer review of "Biomarkers in IBD: What to Utilize for the Diagnosis?"

_diagnostics, 2023, doi:10.3390/diagnostics13182931_

Round 1

Reviewer 1 Report

This review demonstrated that traditional biochemical tests remain helpful in guiding strategies for the diagnosis of IBD. Fecal calprotectin determination is useful to rule out the presence of intestinal inflammation and to avoid unnecessary invasive procedures. New potential indices are promising but at the moment the accuracy for diagnosing ulcerative colitis or Crohn disease is suboptimal and not ready for use in clinical practice. This is an interesting topic and the manuscript was in general well organized that in principle support their conclusions. However, I have some minor comments below:

1.      The authors are suggested to draw a picture for “serological markers and fecal markers”, which should be easy for readers to understand quickly.

2.      The limitation of the existing biomarkers should be discussed.

Author Response

Thank you for your review and for the opportunity to improve our work. In the attached file we provide a revised version of our manuscript, taking into account your suggestions. In particular, some sections have been added, including a new table. You can track changes in bold text.

Reviewer 2 Report

Thank you very much for the opportunity to review the article „Biomarkers in IBD: what to utilize for the diagnosis?”. The article is very interesting, concerning traditional biochemical tests which remain helpful in guiding strategies for the diagnosis of IBD.

I have several remarks.

Routine blood tests are well decsribed. Although, I think the role of CRP in IBD needs more attention. Please, think about descibing it in more details.

Cytokines’ part is very poor. In my opinion there is much more to say about other interleukines, such as IL-1 family, IL-10. Consider decribing them also.

Fecal markers are well decribed. There is no need to change anything.

Thanks.

Author Response

Thank you for your review and for the opportunity to improve our work. In the attached file we provide a revised version of our manuscript, taking into account your suggestions. In particular, some sections have been added, including more information on cytokines. You can track changes in bold text.